# Evaluation of the In Vitro Oral Wound Healing Effects of Pomegranate (*Punica granatum*) Rind Extract and Punicalagin, in Combination with Zn (II)

**DOI:** 10.3390/biom10091234

**Published:** 2020-08-25

**Authors:** Vildan Celiksoy, Rachael L. Moses, Alastair J. Sloan, Ryan Moseley, Charles M. Heard

**Affiliations:** 1School of Pharmacy and Pharmaceutical Sciences, Cardiff University, Cardiff CF10 3NB, UK; celiksoyv@cardiff.ac.uk; 2Oral and Biomedical Sciences, School of Dentistry, Cardiff University, Cardiff CF14 4XY, UK; mosesr@cardiff.ac.uk; 3Melbourne Dental School, Faculty of Medicine, Dentistry and Health Sciences, University of Melbourne, Melbourne, VIC 3010, Australia; alastair.sloan@unimelb.edu.au

**Keywords:** pomegranate, punicalagin, tannins, gingiva, fibroblasts, antioxidant, wound healing

## Abstract

Pomegranate (*Punica granatum*) is a well-established folklore medicine, demonstrating benefits in treating numerous conditions partly due to its antimicrobial and anti-inflammatory properties. Such desirable medicinal capabilities are attributed to a high hydrolysable tannin content, especially punicalagin. However, few studies have evaluated the abilities of pomegranate to promote oral healing, during situations such as periodontal disease or trauma. Therefore, this study evaluated the antioxidant and in vitro gingival wound healing effects of pomegranate rind extract (PRE) and punicalagin, alone and in combination with Zn (II). In vitro antioxidant activities were studied using DPPH and ABTS assays, with total PRE phenolic content measured by Folin–Ciocalteu assay. PRE, punicalagin and Zn (II) combination effects on human gingival fibroblast viability/proliferation and migration were investigated by MTT assay and scratch wounds, respectively. Punicalagin demonstrated superior antioxidant capacities to PRE, although Zn (II) exerted no additional influences. PRE, punicalagin and Zn (II) reduced gingival fibroblast viability and migration at high concentrations, but retained viability at lower concentrations without Zn (II). Fibroblast speed and distance travelled during migration were also enhanced by punicalagin with Zn (II) at low concentrations. Therefore, punicalagin in combination with Zn (II) may promote certain anti-inflammatory and fibroblast responses to aid oral healing.

## 1. Introduction

Wound healing is a complex process, involving a chain of well-orchestrated biochemical and cellular events that effect the repair of diseased or damaged tissues. Healing is mainly achieved through four precise and programmed phases: homeostasis, inflammation, proliferation and remodeling. These phases must occur in an orderly and suitable timeframe which is essential to normal healing, although disruption to these mechanisms by various factors may cause delayed or non-healing to occur [1].

Wounds within the oral cavity can be caused by many factors, including trauma, periodontal disease and surgery. Although the oral cavity harbors a wide variety of commensal microbial species [2], its distinct environment with continuous salivary flow helps prevent contamination and infection [3,4]. Furthermore, although oral and dermal wounds proceed through similar stages of healing, oral wounds are characterized by rapid healing with minimal scar formation, mediated in part via enhanced fibroblast and keratinocyte repair responses [5,6]. However, despite such superior healing properties, oral wounds are common and yet difficult to protect using conventional wound dressing approaches; and therefore are susceptible to microbial contamination and further trauma, such as during mastication [7]. For the treatment of the oral wounds, antibiotics, corticosteroids, non-steroidal anti-inflammatory drugs (NSAIDs) and disinfectants, such as chlorhexidine, have all been used to accelerate the healing process and prevent patient disconformity [8]. However, these drugs are commonly associated with various side effects, such as gastrointestinal damage, discoloration, dysgeusia and excessive sensitivity in the oral mucosa [9]. Therefore, alternative pharmaceutical therapies are needed for the promotion of oral wound healing, which overcome these issues. Natural compounds and formulations may offer such promise, as many have had strong historical roles in the treatment of many different diseases and conditions worldwide. As a result, natural products are favored by modern societies and consumer acceptance is high [10,11]. Indeed, it has been suggested that medicinal plants have more efficacious healing properties and less adverse effects than the other more synthetic pharmaceutical chemicals [12,13].

Pomegranate (*Punica granatum*), a part of the *punicaceae* family native to the Middle East, is a well-established folklore medicine, and mainly cultivated in Iran, India, USA and most of the near and far eastern countries. It has been used in the treatment of dysentery, diarrhoea and stomatitis in traditional medicine in many cultures which is documented in Egyptian Papyrus of Ebers [14,15]. Recent studies have shown that pomegranate demonstrates benefits in treating numerous conditions, due to its anticancer, antimicrobial, anti-inflammatory and antioxidant properties [16]. The different parts of the pomegranate have rich sources of secondary metabolites with potential biological activities [17]. The fruit exocarp (rind) is particularly abundant in hydrolysable tannins, in particular punicalagin, which is a large (mw 1,084.71 g/mol) molecule comprised of gallagic acid and ellagic acid linked via a glucose moiety (Figure 1) [18,19]. These compounds have been attributed as being the primary sources of bioactivity responsible for the desirable medicinal properties of pomegranate, including their dermal wound healing efficacies [19,20,21,22,23,24]. Indeed, from our previous work, pomegranate rind extract (PRE) and punicalagin itself have been shown to exhibit potent anti-inflammatory, antimicrobial and antiviral activities, which can be further potentiated by combination with Zn (II) ions [25,26,27,28]. Zn (II) itself also has a prominent role in all stages of wound repair, regulating immuno-inflammatory cell, endothelial cell, keratinocyte and fibroblast responses [29,30]. Indeed, the importance of Zn (II) to successful wound repair outcomes is supported by studies correlating delayed healing with deficient Zn (II) levels and enhanced repair following the topical application of Zn-containing compounds. Thus, it may be hypothesized that PRE and punicalagin supplementation with Zn (II) can promote additional beneficial wound healing effects. However, whereas beneficial PRE and punicalagin effects on dermal wound healing are supported in the literature, no studies have to date examined whether PRE and punicalagin could offer similar therapeutic wound healing benefits within the oral cavity. Therefore, the purpose of the present study was to evaluate the potential of PRE and punicalagin, with and without Zn (II), used to promote the healing of oral wounds caused by periodontal disease or trauma. Specifically, PRE, punicalagin, Zn (II) alone and Zn (II) in combination with PRE and punicalagin were assessed for their in vitro antioxidant activities, in addition to their effects on the viability, proliferation and migration of human primary gingival fibroblasts.

## 2. Materials and Methods

### 2.1. Materials

Pomegranates were obtained from a local supermarket and were of Spanish origin. Punicalagin (≥98%), [3-(4,5-dimethyl-2-thiazolyl)-2,5-diphenyltetrazolium bromide] (MTT), 2,2′-azino-bis(3-ethylbenzothiazoline-6-sulfonic acid diammonium salt (ABTS), 2,2-diphenyl-1-picrylhydrazyl (DPPH), Folin–Ciocalteu (F-C) reagent, potassium persulphate, (±)-6-hydroxy-2,5,7,8-tetramethylchromane-2-carboxylic acid (Trolox); dimethylsulfoxide (DMSO), ascorbic acid and sodium carbonate (Na_2_CO_3_) were all obtained from Sigma-Aldrich (Gillingham, UK). Zinc sulfate heptahydrate (ZnSO_4_·7H_2_O), potassium hydrogen phthalate, Dulbecco’s Modified Eagle Medium (DMEM), fetal calf serum (FCS), L-glutamine and antibiotics/antimycotics were obtained from ThermoFisher Scientific (Loughborough, UK).

### 2.2. Preparation of Pomegranate Rind Extract (PRE)

The rind of the pomegranates was peeled with a scalpel and cut to approximately 2 cm^2^ pieces. The net weight of the rind was 300 g. This was blended (25%) *w*/*v* in deionized water in a standard blender until visibly homogenous. The blended rind in deionized water was boiled for 10 min and centrifuged (×4) using a Heraeus Multifuge 3S/3S-R centrifuge (5980× *g* at 4 °C for 30 min), before filtration through a Whatman 0.45 µm nylon membrane filter. The collected solution was freeze dried, protected from light and stored at −20 °C until required. The desired concentration of PRE was prepared in pH 4.5 phthalate buffer and sterilized by using a 0.45 μm Millex-FG syringe-driven filter [26]. Punicalagin concentrations were determined by HPLC (Thermo LCQ classic LCMS with ESI source) according to method of Seeram et al. [31]; and found that 1 mg/mL PRE contains approximately 17 μg punicalagin.

### 2.3. Determination of Total Phenolic Content

The Folin–Ciocalteu (F-C) colorimetric assay was used to quantify the total phenolic content in PRE, according to method by Ainsworth and Gillespie [32]. Briefly, 0.5 mg/mL PRE samples were prepared and 200 μL 10% (*v*/*v*) F-C reagent was added to 100 μL of the prepared PRE samples, followed by the addition of 800 μL 700 mM Na_2_CO_3_. Samples were incubated at room temperature for 2 h. After the incubation period, 200 μL of each sample was added to 96-well plates and the absorbance values read at 760 nm on a plate reader (Fluostar Optima, BMG Labtech, Aylesbury, UK). The concentration of phenolic compounds in the PRE was shown as tannic acid equivalents (TAE) per gram of freeze-dried sample.

### 2.4. The 2,2-diphenyl-1-picrylhydrazyl (DPPH) Radical Scavenging Assay

The DPPH assay was used to evaluate the scavenging of stable radicals by PRE, punicalagin, Zn (II) and PRE and punicalagin in combination with Zn (II), as previously described [33]. Briefly, samples were initially prepared in 0.2 mM DPPH solution and two-fold serial dilutions made in 96-well plates for each sample. Plates were wrapped in foil and incubated for 30 min at room temperature. After 30 min, the absorbance values of each sample were read at 515 nm as above, versus samples containing only DPPH (negative control), with ascorbic acid used as a positive control. The % of the radical scavenging activities of each sample was calculated as follows:% DPPH scavenging = 100 × [1 − (OD_sample_/OD_control_)

The concentration of each sample which scavenged 50% of the initial DPPH radicals generated was calculated by interpolating the [(Abs of the sample) − (Abs sample blank)] into a calibration curve generated by the DPPH absorbance values at different sample concentrations. All assays were performed on 3 separate occasions, with each experiment including 3 replicates.

### 2.5. The 2,2’-azino-bis(3-ethylbenzothiazoline-6-sulfonic-acid (ABTS) Radical Scavenging Assay/Trolox Equivalent Antioxidant Activity (TEAC)

The antioxidant potential of PRE, punicalagin, Zn (II) and PRE and punicalagin in combination with Zn (II), was also assessed using the ABTS/TEAC assay, based on the study by Re et al. [34]. This assay is based on the ability of compounds to scavenge the ABTS radical, produced by the reaction between 7 mM ABTS and 2.45 mM potassium persulphate. The ABTS solution was prepared and diluted to a final absorbance of 0.7 ± 0.2 at 734 nm obtained using a plate reader, as described above. The antioxidant capacities of PRE and punicalagin (both 0.5 mg/mL) and 0.1 mM Zn (II) were determined. Trolox (0–400 μg/mL) was used as a positive control and to express the data as Trolox equivalent antioxidant capacity (TEAC). All assays were performed on 3 separate occasions, with each experiment including 3 replicates.

### 2.6. Cell Culture

Human primary gingival fibroblasts were obtained from the American Type Cell Culture Collection (ATCC, Manassas, VA, USA). Gingival fibroblasts were cultured in DMEM supplemented with 10% heat-inactivated FCS, 1% L-glutamine (2 mM) and 1% antibiotic/antimycotic solution. Cells were incubated at 37 °C in a humidified atmosphere of 5 % CO_2_. The passage number of cells used in all experiments was between 2 and 7.

Samples were prepared fresh on the day of treatment. Different concentrations of PRE (0.1–100 μg/mL), punicalagin (0.1–10 μg/mL), ZnSO_4_·7H_2_O (0.1 mM) and PRE and ZnSO_4_·7H_2_O (0.1 mM), punicalagin and ZnSO_4_.7H_2_O (0.1 mM) were prepared. Freeze-dried PRE, Zn (II) and punicalagin were firstly dissolved in phthalate buffer pH 4.5 to make the stock solutions, and then filtered using 0.2 µm Minisart syringe filters made of acrylic resin, methacrylate butadiene styrene (Sartorius Stedim Biotech GmbH, Göttingen, Germany), under sterile conditions. Compound concentrations were further prepared in DMEM containing 1% FCS, 1% L-glutamine and 1% antibiotics/antimycotics. Control culture medium was also supplemented with 1% phthalate buffer pH 4.5 to negate any influences on cellular activities by the buffer itself.

### 2.7. Cell Viability and Proliferation

The effects of PRE, punicalagin, Zn (II) and PRE and punicalagin in combination with Zn (II) on gingival fibroblast viability and proliferation were determined MTT assay [35]. Gingival fibroblasts were seeded into 96-well plates at a density of 2.5 × 10^3^ cells/well and cultured at 37 °C/5% CO_2_ for 24 h. After 24 h, the media was changed to serum-free DMEM and the cells cultured for a further 24 h. Cells were subsequently treated with various concentrations of the samples for 24, 48 and 72 h, with media changes every 24 h. At each time point, 25 μL MTT (5 mg/mL in phosphate buffered saline, PBS) was added to each well and cultured at 37 °C/5% CO_2_ for 4 h. After 4 h incubation, the MTT was discarded and each well treated with 100 μL pure DMSO, followed by further incubation at 37 °C/5% CO_2_ for 30 min, with light protection. The absorbance values of each well were then read at 570 nm. Sample effects on cell viability and proliferation were expressed as % viable cells versus untreated controls, which were arbitrarily assigned a viability of 100%. All assays were performed on 3 separate occasions, with each experiment including 6 replicates.

### 2.8. Cell Migration and Wound Repopulation

The effects of PRE, punicalagin, Zn (II) and PRE and punicalagin in combination with Zn (II), on fibroblast migration were assessed for the ability to stimulate in vitro scratch wound repopulation, as previously described [36]. Gingival fibroblasts were seeded into 24-well plates at a density of 2.5 × 10^4^ cells/well and cultured at 37 °C/5% CO_2_ for 48 h. After 48 h, the media was changed to serum-free DMEM and the cells cultured for a further 24 h. Serum-free DMEM was removed and scratch wounds made using sterile pipettes. Fibroblasts were subsequently treated with different concentrations of test sample, with untreated cells in serum-free media serving as negative controls. Cell migration and wound repopulation were monitored by automated time-lapse microscopy, using a Cell-IQ^®^ Automated Cell Culture and Analysis System (Chip-Man Technologies Ltd., Tampere, Finland), at 37 °C/5% CO_2_. Digital images taken every 20 min over a 48 h period, using Cell-IQ Analyser™ Software, whilst ImageJ^®^ Software (Version 1.49, https://imagej.nih.gov/ij/), were used to quantify cell migration parameters, including: cell displacement (Td), overall velocity (Td/t), distance travelled (Tt) and migratory speed (Tt/t). Each experiment was performed on 3 separate occasions, with each experiment including 3 replicates.

### 2.9. Statistical Analysis

Data values were expressed as the average ± standard error of the mean (SEM). Statistical analysis of antioxidant data was performed using the Duncan’s multiple range test. Statistical analysis of gingival fibroblast viability, proliferation and migration was performed by one-way ANOVA with Tukey’s multiple comparison post-test. Statistical analyses were performed using the GraphPad Prism, Version 8.00 (GraphPad Software, San Diego, CA, USA). Significance was considered at *p* < 0.05.

## 3. Results

### 3.1. Total Phenolic Content

The quantitative determination of the total phenolic content of PRE was expressed in μg TAE per g of freeze-dried PRE. The results showed that, on average, freeze-dried PRE contained 496 mg TAE/g.

### 3.2. Antioxidant Activities Using DPPH and ABTS Assays

The antioxidant capacities of PRE, punicalagin and their combination with 0.1 mM Zn (II) were assessed using both the DPPH and ABTS assays (Figure 2). The results of the DPPH assay are expressed as % of DPPH inhibition and IC_50_ values (the sample concentration needed to inhibit 50% of the initial DPPH free radical flux). All samples studied showed a dose-dependent response in the % of free DPPH inhibition. PRE (10.69 ± 0.44%) and PRE in combination with 0.1 mM Zn (II) (8.1 ± 0.27%) were required at higher concentrations than the ascorbic acid positive controls (8.31 ± 0.64%), to induce 50% inhibition. However, punicalagin (6.04 ± 0.29%) and its combination with 0.1 mM Zn (II) (6.99 ± 0.20%) required lower concentrations than the ascorbic acid controls. While there was a slight difference between the compounds and their Zn (II) combinations, no statistically significant differences were observed (*p* > 0.05). The ABTS assay showed similar patterns of antioxidant capabilities to the DPPH assay. Punicalagin and punicalagin with 0.1 mM Zn (II) exhibited significantly higher TEAC values than PRE and PRE with 0.1 mM Zn (II) (*p* < 0.001). Similarly, 0.1 mM Zn (II) addition did not cause any significant change in the antioxidant scavenging activities of PRE or punicalagin (*p* > 0.05).

### 3.3. Effects on Gingival Fibroblast Viability and Proliferation

The effects of PRE, punicalagin, Zn (II) and PRE and punicalagin in combination with Zn (II) on fibroblast viability and proliferation, were determined by MTT assay (Figure 3). According to the data obtained, 1 mM Zn (II) significantly reduced fibroblast viability and proliferation at all time- points (*p* < 0.001), while lower Zn (II) concentrations did not exhibit such decreases at 24 or 48 h (*p* > 0.05). When fibroblasts were treated with PRE or punicalagin alone, both showed dose-dependent decreases in cell viability from 24 h onwards, at the highest concentrations of PRE (100 μg/mL) and punicalagin (10 μg/mL) examined (*p* < 0.001). In contrast, lower concentrations of neither PRE nor punicalagin influenced cell viability, versus untreated negative controls (NC, *p* > 0.05). However, PRE and punicalagin combined with 0.1 mM Zn (II) significantly reduced fibroblast viability (*p* < 0.001), although treatment with 0.1 mM Zn (II) alone did not decrease viability (*p* > 0.05).

### 3.4. Effects on Gingival Fibroblast Migration and Wound Repopulation

The effects of PRE, punicalagin, Zn (II) and PRE and punicalagin in combination with Zn (II) on fibroblast migration and wound repopulation were evaluated using the in vitro scratch wound assays, with cell displacement (Td), overall velocity (Td/t), distance travelled (Tt) and the migratory speed (Tt/t) of the gingival fibroblasts monitored. PRE (100 μg/mL) and punicalagin (10 μg/mL) reduced gingival fibroblast migration and wound repopulation, significantly decreasing fibroblast speed compared to untreated controls (*p* < 0.001, Figure 4 and Figure 5). In contrast, lower PRE and punicalagin concentrations increased the speed, cell displacement, overall velocity and distance travelled. However, no significant differences in these cellular parameters were determined versus untreated controls (all *p* > 0.05).

Fibroblasts treated with 0.1 mM Zn (II) did not show any significant differences compared to untreated controls (*p* > 0.05, Figure 4 and Figure 6). However, the combination of 0.1 μg/mL punicalagin and 0.1 mM Zn (II) induced a significant increase in cell speed and distance travelled, versus untreated controls (*p* < 0.001). Likewise, although there was a significant decrease in fibroblasts treated only with punicalagin at the highest concentration (10 μg/mL), when combined with 0.1 mM Zn (II), this inhibitory effect was not observed (*p* > 0.05).

## 4. Discussion

In light of its folklore medicinal status, crude pomegranate extracts and its constituent compounds, such as punicalagin, have received much biomedical attention given the considerable evidence supporting their efficacy against a wide range of diseases and conditions, ascribed to its various anticancer, antimicrobial, anti-inflammatory and antioxidant bioactivities [16]. Although numerous studies have previously endorsed the beneficial effects of PRE and punicalagin and advocated their application in the treatment of impaired wound healing responses in skin [19,20,21,22,23,24], a clinical area which has largely been overlook from a wound healing viewpoint are the potential abilities of PRE and punicalagin within the oral cavity, when tissue damage is commonly caused by periodontal disease and trauma. Indeed, periodontal diseases, comprising gingivitis and periodontitis, are regarded as being the most common disease of mankind, leading to huge economic burdens for healthcare providers worldwide [37]. As prevalence is also associated with risk factors such as age and diabetes, projections estimate further escalations in incidence with ever-increasing age demographics and diabetic rates worldwide. Although a wide array of therapeutic entities are available, these predominantly possess antibiotic, antimicrobial or anti-inflammatory properties, thereby indirectly promoting periodontal healing through the eradication of dental plaque/bacterial biofilm accumulation and/or the exacerbation of chronic inflammatory responses [8,37]. Furthermore, despite the development of a plethora of antibiotic and non-antibiotic-based drug delivery approaches to counteract microbial accumulation, biofilm formation or the inflammation associated with periodontal disease, few agents have fully progressed to routine clinical use [38,39]. Thus, in addition to addressing the side effects commonly associated with such therapeutics, the development of efficacious pharmaceutical options with established potent antimicrobial, anti-inflammatory and pro-healing properties, such as pomegranate, could meet a significant clinical and public health need in reducing the prevalence and severity of such conditions on a global scale.

The bioactivities of pomegranate extracts are generally attributed to its phenolic contents. Although the whole fruit comprises a large number of phenolic compounds, including anthocyanins, gallotannins, hydroxycinnamic acids, hydroxybenzoic acids and hydrolysable tannins. Compared to other parts of the fruit, pomegranate rind is known to contain the highest levels of bioactive polyphenolics, especially hydrolysable tannins such as punicalagin, which are responsible for the antioxidant activities of the PRE [18,40,41,42,43]. Indeed, potent antioxidant activity could play an important role in periodontal wound healing, as chronic inflammation, excessive reactive oxygen species (ROS) production and oxidative stress are key contributors to the host connective tissue damage associated with periodontal disease pathology [44,45]. In this study, it was shown that the total polyphenol content of the aqueous extract of PRE was 496 mg TAE/g freeze dried pomegranate rind. This result is similar to study by Malviya and Jha [46], who quantified the total polyphenol content of pomegranate rind using different solvents and found that water extract had the highest value, 435 mg TAE/g pomegranate rind. Furthermore, in this study, the antioxidant activity was evaluated using DPPH and ABTS assays. In both assays, punicalagin showed significantly higher antioxidant activity than PRE when at the same concentration as punicalagin, although addition of Zn (II) did not cause any significant changes in the ROS scavenging capacities of punicalagin and PRE, probably due to the stability of the Zn (II) ion in respect of redox reactions. Seeram et al. [47] found that pomegranate juice had higher antioxidant activity than punicalagin when they used the same concentrations of pomegranate juice and punicalagin and suggested a synergistic/additive activity of polyphenols than only one compound for this result. However, it is very difficult to assess the antioxidant activity, using a single method, since it can provide only basic information about antioxidant activity but using different methods can give more detail. There could be differences between the results because of extract and sample preparation, selection of endpoints and expression of results [48]. That said, it has been suggested that there is a correlation between the phenolic contents and antioxidant properties of PRE, with the most abundant polyphenol being punicalagin [46,49,50]. Thus, as the antioxidant assay data in this study provided a dose-dependent response, this may further imply that polyphenols are responsible for the antioxidant activity in PRE and punicalagin. Indeed, punicalagin showed higher antioxidant activity than PRE at the same concentrations in both DPPH and ABTS assays, as PRE contains a range of non-phenolic compounds. Therefore, it may be suggested that the antioxidant activity of PRE could be attributed to its punicalagin content, in line with previous findings [18,40,41,42,43,50].

Assessment of PRE and punicalagin effects on human gingival fibroblasts alone and in combination with Zn (II) showed no stimulation of fibroblast proliferation over the 72 h culture period. In contrast, PRE and punicalagin significantly reduced fibroblast viability at high concentrations (100 μg/mL and 10 μg/mL respectively) and when applied with Zn (II). Similar findings have been reported with other natural compounds and extracts, such as propolis, where despite its antimicrobial and antioxidant properties mediated through its polyphenol constituents, it can promote significant fibroblast cytotoxicity when co-administered with Zn (II) [51]. Furthermore, numerous studies have demonstrated the anti-proliferative or cytotoxic activities of PRE and punicalagin against a wide range of cancer cell types [22,52,53,54,55] and fibroblasts [56]. Similarly, although many studies have shown the stimulatory effects of Zn (II) on keratinocyte proliferation [29,31], negligible or inhibitory effects on fibroblast proliferative responses have been identified for Zn (II) [57,58]. However, such responses are likely to be concentration dependent, as fibroblasts are reported as being resistant to Zn (II) cytotoxicity <500 mM [59], as evident here.

Further studies evaluated PRE and punicalagin effects on gingival fibroblast migration and wound repopulation alone and in combination with Zn (II), via the analysis of relevant parameters including cell migration speed, cell displacement, overall velocity and distance travelled, over 48 h in culture. High concentrations of PRE (100 μg/mL) and punicalagin (10 μg/mL) significantly inhibited fibroblast migration and wound repopulation, presumably as a consequence of the cytotoxic effects identified above. However, lower PRE and punicalagin concentrations maintained or enhanced fibroblast migration and wound repopulation, equivalent to untreated controls. Furthermore, despite Zn (II) alone exerting no effects on fibroblast wound repopulation, 0.1 μg/mL punicalagin combined with 0.1 mM Zn (II) induced significant increases in cell speed and distance travelled, versus untreated controls; whilst 0.1 mM Zn (II) supplementation also attenuated the inhibitory effects of punicalagin (10 μg/mL) on cell speed. Such stimulatory effects on cell migration are significant, as previous studies have mostly described the inhibitory effects of PRE and punicalagin on cell motility/invasion, for instance, in cancer cells [22,52,53]. However, as Zn (II) and Zn-containing compounds can significantly enhance fibroblast migration and wound closure in vitro and in vivo [29,31,60], Zn (II) may actually be the key mediator of the increased fibroblast migratory responses identified herein.

Fibroblasts play a pivotal role in mediating wound healing responses, from initial cellular migration, proliferation and cytokine/growth factor production to subsequent extracellular matrix (ECM) synthesis/remodeling, wound contraction and closure. Thus, as wound repopulation is acknowledged to be dependent on the induction of both migratory and proliferative responses [61], the data presented herein would suggest that punicalagin and Zn (II) primarily promote oral fibroblast migration, rather than proliferation, in light of the absence of stimulated fibroblast proliferative responses induced by these concentrations alone or in combination. However, although oral and dermal wounds proceed through similar stages of healing, oral wounds are well- characterized by minimal inflammatory and angiogenic responses, rapid healing and minimal scar formation; unlike dermal wounds [5,6], with such superior healing responses attributed to the specialized genotypic and phenotypic properties of fibroblasts residing within oral tissues. In contrast, in non-healing skin wounds, such cellular responses are impaired, leading to failed wound closure [62]. Thus, fibroblast viability and induced proliferative and migratory responses are key events in normal repair processes, although differences in the responses of oral and dermal fibroblasts to specific PRE or punicalagin concentrations may be a consequence of the well-established differences in proliferative and migratory capabilities which exist between these distinct fibroblast populations [6].

Although the positive dermal wound healing activities of PRE and punicalagin have been recognized for some time [19,20,21,22,23,24], our findings are the first to report on the potential wound healing benefits of punicalagin in combination with Zn (II) to oral wounds caused by periodontal disease or trauma, in terms of alleviating ROS levels and oxidative stress and by stimulating gingival fibroblast migration. Whereas such particular antioxidant and pro-migratory responses could benefit gingival repair processes, it remains to be determined whether PRE or punicalagin alone or with Zn (II) supplementation possess any bactericidal, bacteriostatic or anti-biofilm properties versus the pathogenic Gram-negative bacterial species commonly associated with the initiation and progression of periodontal disease, such as *Porphyromonas gingivalis* [63], as established with microflora from other clinical situations [16,25,40,46]. However, such antimicrobial properties are currently under investigation. As uncontrolled biofilms initiate and sustain the inflammatory and resident connective tissue cell destruction in periodontal tissues, the therapeutic limitation or eradication of microbial biofilm accumulation by PRE or punicalagin would undoubtedly help inhibit the development and progression of periodontal disease evoking further tissue reparative responses.

## 5. Conclusions

Although pomegranate (*Punica granatum*) extracts and its bioactive constituents, such as punicalagin, have been used since ancient times to treat a broad range of diseases and conditions, only now have studies begun to assess its therapeutic potential for the treatment of wounds within the oral cavity, such as those manifested during periodontal disease or trauma. Both PRE and punicalagin were shown to possess potent antioxidant capabilities, whilst punicalagin combined with Zn (II) further induced human gingival fibroblast migration and wound repopulation responses, but exerted no stimulatory effects on fibroblast proliferation. Therefore, purified punicalagin in combination with Zn (II) may offer potential benefits as a natural compound-based therapy, aiding wound healing mechanisms within the oral cavity.

## Figures and Tables

**Figure 1 biomolecules-10-01234-f001:**
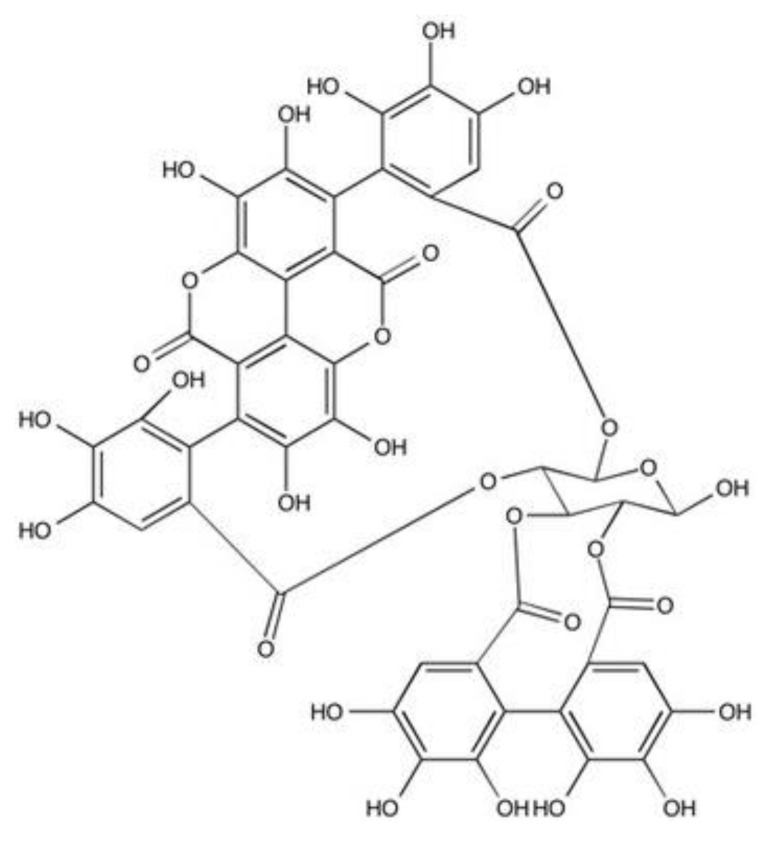
The chemical structure of punicalagin.

**Figure 2 biomolecules-10-01234-f002:**
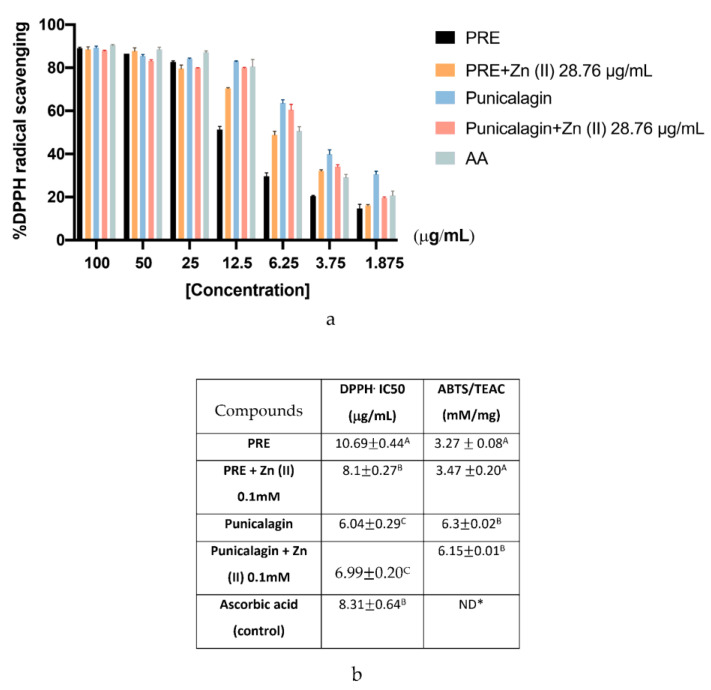
Antioxidant capabilities of PRE and punicalagin alone and in combination with 28.76 μg/mL (0.1 mM) Zn (II). (**a**) % DPPH antioxidant scavenging capacities at different sample concentrations. (**b**) TEAC values obtained for each sample, based on the finding of the ABTS assay. Values are presented as the mean ± SEM (*n* = 3). TEAC, Trolox equivalent antioxidant capacity. Values followed by the same capital letter within the same column are not significantly different *(**p* > 0.05) between the compounds analyzed by Duncan’s multiple range test. * ND; not determined.

**Figure 3 biomolecules-10-01234-f003:**
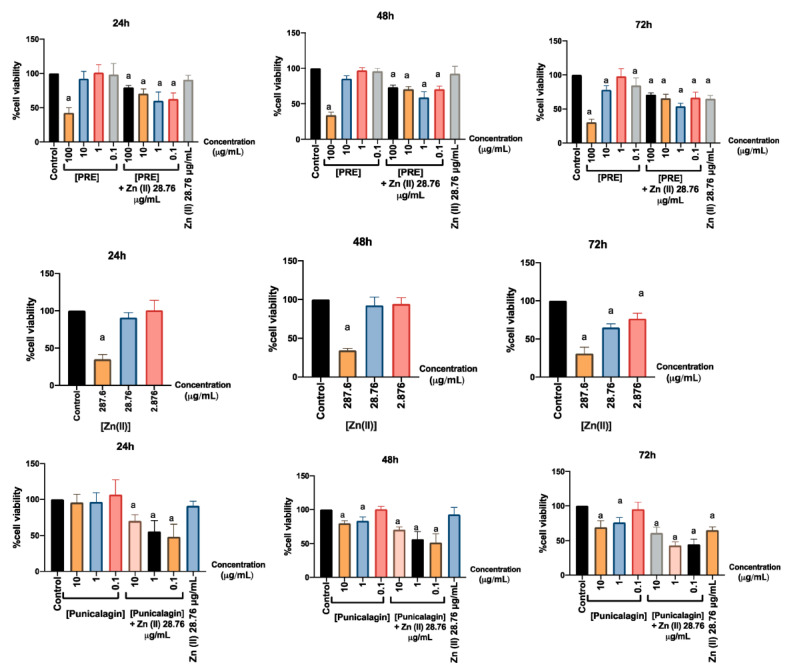
Effects of PRE (0.1–100 μg/mL), punicalagin (0.1–10 μg/mL) and 28.76 μg/mL (0.1 mM) Zn (II) alone and in combination with Zn (II), on human gingival fibroblast viability and proliferation at 24, 48 and 72 h, as determined by MTT assay. Values are presented as the mean % ± SEM (*n* = 3). Mean values with an “a” letter was significantly different than the untreated negative controls (*p* < 0.001).

**Figure 4 biomolecules-10-01234-f004:**
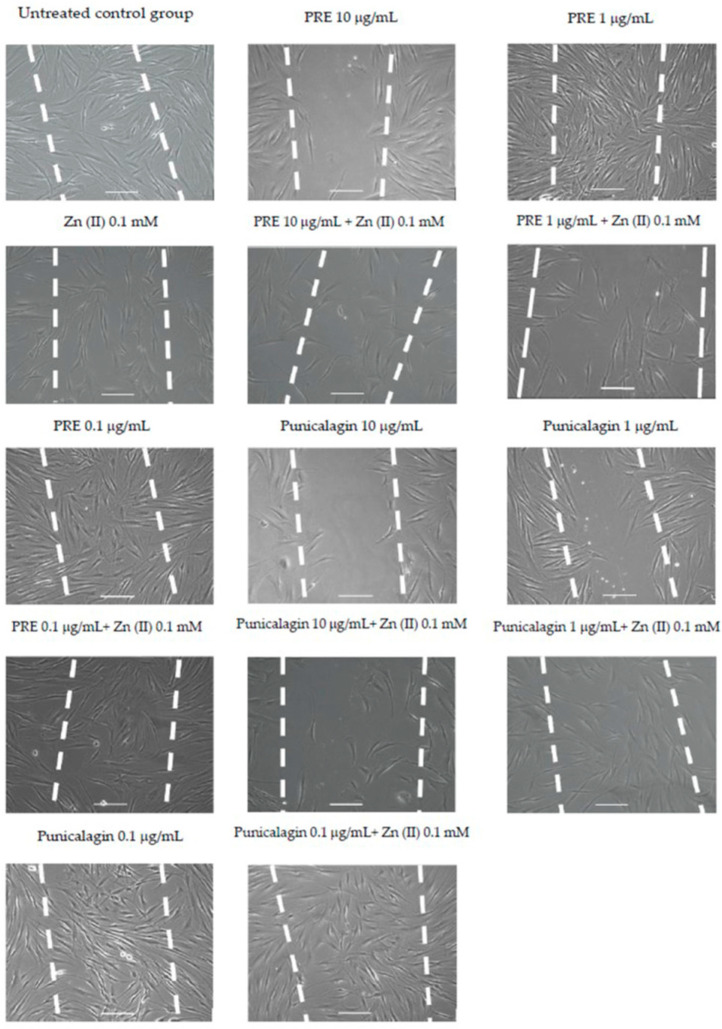
Representative time-lapse microscopy images of gingival fibroblast migration and wound repopulation at 48 h, following treatment with PRE and punicalagin (0.1–10 μg/mL) alone and with 0.1 mM Zn (II), White dashed lines show original scratch wounds at 0 h. Scale bar = 100 μm.

**Figure 5 biomolecules-10-01234-f005:**
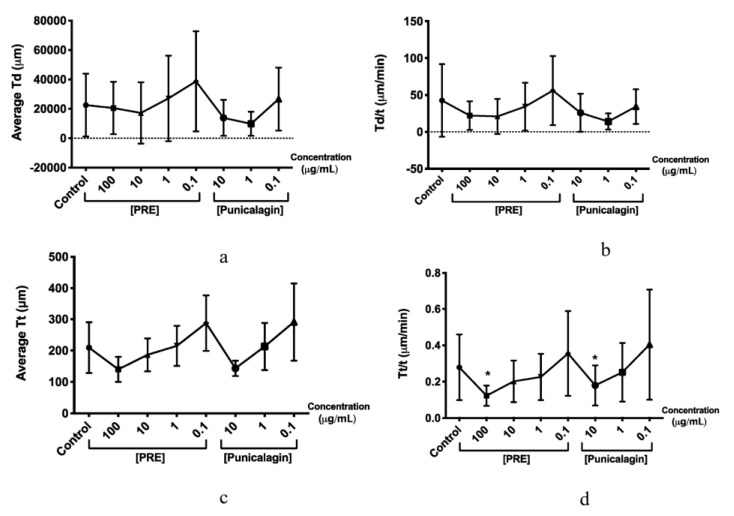
Effects of PRE (0.1–100 μg/mL) and punicalagin (0.1–10 μg/mL) on human gingival fibroblast scratch wound migration parameters, over 48 h. (**a**) Cell displacement (Td), (**b**) overall velocity (Td/t), (**c**) distance travelled (Tt), and (**d**) migration speed (Tt/t). Values are presented as the mean ± SEM (*n* = 3, * *p* < 0.05).

**Figure 6 biomolecules-10-01234-f006:**
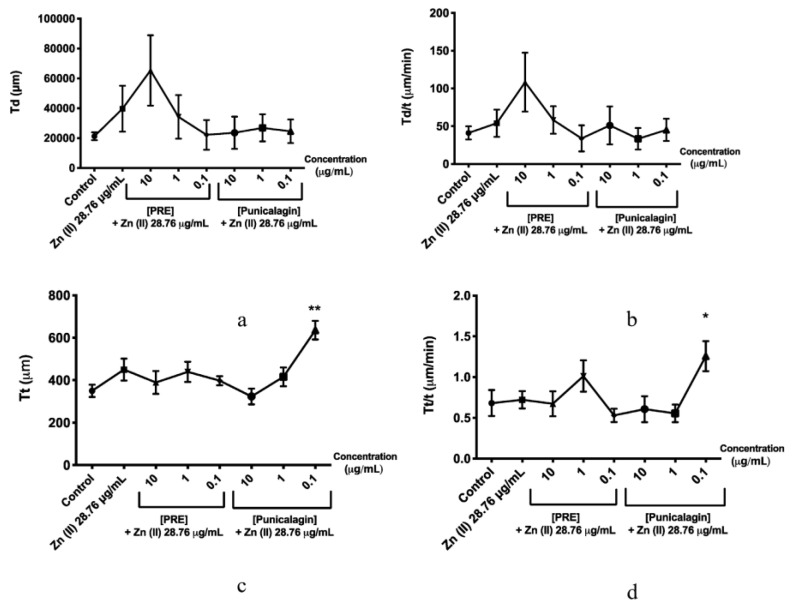
Effects of PRE (0.1–10 μg/mL) and punicalagin (0.1–10 μg/mL) in combination with 28.76 μg/mL (0.1 mM) Zn (II), on human gingival fibroblast scratch wound migration parameters, over 48 h. (**a**) Cell displacement (Td), (**b**) overall velocity (Td/t), (**c**) distance travelled (Tt), and (**d**) migration speed (Tt/t). Values are presented as the mean ± SEM (*n* = 3, * *p* < 0.05, ** *p* < 0.01).

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
