# Peer review of "Evaluation of the In Vitro Oral Wound Healing Effects of Pomegranate (Punica granatum) Rind Extract and Punicalagin, in Combination with Zn (II)"

_biomolecules, 2020, doi:10.3390/biom10091234_

Round 1
Reviewer 1 Report
An interesting manuscript outlining some original research which I enjoyed reading. I believe it is suitable for publication in Biomolecules with the following amendments.
The principal area where I feel more context is required is the fruit itself - it is predominantly known as a fruit/foodstuff not a “folklore medicine”. For example, where is the pomegranate predominantly cultivated? In which countries/ethnic groups is the pomegranate typically used as a traditional medicine? This information is important to ensure that any future successful medicinal use of pomegranate or its derivatives is (partly) attributable to its historic use a particular country/ ethnic group.
The submission can be improved by also addressing the following points:
Line 54 – “bacterial resistance” is not a side-effect.
Line 108 – what procedure was used for the blending? Please state.
Line 170 – from what material were the syringe filters made?
Line 262 – the following sentence is unclear: “Data with different uppercase superscript letters in the same column indicate the significant difference (p<0.05) between the compounds analysed by Duncan’s multiple range test.”
Line 384 – there is no need to include “(exocarp)” as this has been previously stated.
Figure 2 (a) – x-axis units error, also no axis title, e.g. “Concentration”.
Figure 3 (all charts) – x-axis units error, also no axis title, e.g. “Concentration”.
Figure 5 (b), (c) and (d) – x-axis units error, also no axis title, e.g. “Concentration”.
Figure 6 (a), (b) and (d) – x-axis units error, also no axis title, e.g. “Concentration”.
Author Response
Reviewer 1
Main Points
- Although oral and dermal wounds proceed through similar stages of healing, oral wounds are well-characterized by minimal inflammatory and angiogenic responses, rapid healing and minimal scar formation, in contrast to dermal wounds (reviewed by Glim et al., Wound Rep. Regen., 2013). Such superior healing and minimal scarring responses are attributed to the specialised genotypic and phenotypic properties of fibroblasts and keratinocytes residing within oral tissues. Although information to this effect is already presented in the Introduction (highlighted in yellow, page 2), as requested by the Reviewer, further details are also now provided in the Discussion (highlighted in yellow, pages 14-15).
Minor Points
- As requested by the Reviewer, the reference formatting has been update and standardised throughout.
Reviewer 2 Report
Pomegranate (Punica granatum) as well as its rich hydrolysable tannin, punicalagin has been well-characterized for its wound healing effects. In this study, the authors aimed to demonstrate oral wound healing effects of pomegranate rind extract and punicalagin, alone and in combination with Zn (II). The work is technically sound and the results are acceptable. Some suggestion is listed below for the consideration of revision.
Major:
The authors are suggested to describe the differences between dermal wound healing and oral wound healing as well as the different effects of pomegranate rind extract and punicalagin on these two types of wound healing. Please explain or discuss the possible reasons causing the different effects.
Minor:
Formats of some journal names in the references are incorrect.
Author Response
Reviewer 2
Main Points
- As requested by the Reviewer, a brief section has now been included in the Introduction from a historical perspective on the use of pomegranate as a folklore medicine, including some of the ethnic groups who used pomegranate and some of the diseases and conditions pomegranate was used to treat (highlighted in green, page 2).
- As requested by the Reviewer, ‘bacterial resistance’ has now been removed as a side-effect from Line 54 (highlighted in blue, pages 2).
- As requested by the Reviewer, additional information on the blending procedure has now been added to Line 108 (highlighted in yellow, pages 3-4).
- As requested by the Reviewer, additional information on the syringe filter has now been added to Line 170 (highlighted in yellow, page 5).
- As requested by the Reviewer, changes have been made in the figure.
- As requested by the Reviewer, ‘exocarp’ has now been removed from Line 384 (highlighted in yellow, page 13).
- As requested by the Reviewer, changes have been made in the figure.
- As requested by the Reviewer, changes have been made in the figure.
- As requested by the Reviewer, changes have been made in the figure.
- As requested by the Reviewer, changes have been made in the figure.